# Planar Reconstruction of Regular Surfaces' Three-Dimensional Morphology and Tribology Application

**Xiang Xu** [1,2] , **Zekang Feng** [1,2], **Nengqi Xiao** [1,2,3], **Xinze Zhao** [1,2,3,*] **and Zuyue Zhang** [1,2]

[1]  Hubei Key Laboratory of Hydroelectric Machinery Design & Maintenance, China Three Gorges University, Yichang 443002, China; xiang_xu@ctgu.edu.cn (X.X.); fzk13476591574@126.com (Z.F.); xiaonengqi@126.com (N.X.); zuyv_zhang@126.com (Z.Z.)

[2]  Hubei Engineering Research Center for Graphite Additive Manufacturing Technology and Equipment, China Three Gorges University, Yichang 443002, China

[3]  National United Engineering Laboratory for Advanced Bearing Tribology, Henan University of Science and Technology, Luoyang 471023, China

*  Correspondence: xzzhao@ctgu.edu.cn; Tel.: +86-139-7260-3535

**Abstract:** The three-dimensional morphology of frictional sub-surfaces holds significant importance for studying tribological issues. However, the uniformity of the horizontal datum in 3D scanning is limited for curved surfaces, resulting in the inability to obtain accurate contour characterization parameters from the scanning results. This study aims to address this issue by constructing regular surface equations and normalizing the 3D profiler scanning results. By fitting the data, a "plane" surface representative of the surface features is obtained, and the paper demonstrates this approach on the surface morphology of different worn parts in the frictional area of spherical bearings located in the specific environment of the Three Gorges gate. The results indicate that the obtained "plane" effectively reconstructs the three-dimensional morphology map of the regular surface. Moreover, this reconstructed plane not only clearly illustrates the surface characteristics but also provides the foundation for analyzing the wear mechanism.

**Keywords:** regular surface; three-dimensional morphology; plane reconstruction; spherical bearing; wear

## 1. Introduction

Three-dimensional morphology, as a representation of surface geometric features, finds extensive applications in tribological research [1]. Gou [2] experimentally analyzed the friction coefficients, microstructures, wear rates, three-dimensional surface morphology, and friction film transfer behavior of different balls and PDC (spolycrystalline diamond compact). The results show that there are significant differences in the friction and wear behaviors of PDC and different steel balls in the drilling fluid environment. In a study by S. Jithin [3], a comprehensive characterization of three-dimensional morphology was performed on discharge-woven SS304 surfaces. The analysis revealed that as discharge energy increased, the peak count distribution tended to flatten, with an observed shift towards higher heights. C. Gachot [4] employed a surface weaving technique using preheated embossing to obtain a herringbone weave pattern and characterized it using 3D morphology. Mozgovoy [5] analyzed the planar 3D morphology and determined the amount of Al-Si transferred to AlCrN and CrWN surfaces. In order to establish the relationship between process parameters, machined surface texture, and the frictional properties of rotational ultrasonically rolled scale surfaces, Zhao [6] used the three-dimensional surface roughness characteristic parameters $Sa$, $Sz$, and $Sq$ to evaluate the scalelike surface morphology. Zuo [7] investigated the three-dimensional morphology of HNBR planes after testing and discovered that hard particles have a minimal impact on microfabrication. Zhao [8] proposed a model-based method for the numerical simulation and characterization of sliding

wear surface morphology evolution using a deterministic elastohydrodynamic lubrication (EHL) model and a surface 3D morphology fractal method. The validity of the lubrication model was verified through oil film thickness measurements, and the actual surface wear coefficients were obtained from wear experiments. Zhang [9] utilized scanning techniques to obtain the three-dimensional morphology of metal ring specimens and established a correspondence between surface volatility and roughness *Ra*. Jiang [10] introduced a method for accurately characterizing the curved surface morphology of microgrooves produced by micromilling. The method involved extracting the reference plane of the curved surface using the bidimensional empirical mode decomposition algorithm and characterizing the three-dimensional surface roughness thereafter.

However, the aforementioned experiments primarily focused on testing horizontal surfaces or inclined planes with small inclination angles. When it comes to the surface morphology of curved surfaces, the absence of a planar datum makes it challenging to carry out further analysis and comparison of the test data. Consequently, obtaining parameters that can effectively characterize the surface geometry becomes unfeasible. Additionally, during the process of parts processing, dealing with non-regular or complex surfaces not only adds complexity to the process but also presents significant challenges in terms of achieving the desired processing accuracy [11]. As a result, regular surfaces are more commonly used. Therefore, it is of the utmost importance to analyze regular surfaces and reconstruct their three-dimensional morphology in order to obtain specific surface characteristics.

Therefore, this study employs a non-contact three-dimensional profiler [12,13] to scan the regular surface and obtain its initial three-dimensional profile. Based on the scanning data, the vertical depth of the surface between each measurement point and the corresponding theoretical point in the three-dimensional profile is calculated, considering the correspondence between the horizontal position and height of each point. By considering the radial slope of each point, the regular surface is reconstructed as a flat plane, allowing for the observation of surface profile parameters on the spherical surface. Subsequently, the accuracy of the reconstructed results is verified by analyzing the worn surface of a spherical bearing, providing empirical evidence for the acquisition of accurate characteristic parameters of the surface profile.

## 2. Mathematical Analysis

During the process of measuring the three-dimensional shape of an object, it is important to consider that the surface may not be perfectly smooth and may contain abrasive marks. Therefore, a surface reconstruction method accounting for the presence of these abrasive marks needs to be proposed. In this study, we use a regular sphere as an example to explain the idea of plane reconstruction. Similar to differentiation, the scanning range of the sphere is divided into several small sectors, and the reconstructed surface is obtained by "unfolding" these sectors.

When scanning the regular sphere, the probe of the 3D profiler scans the surface in the manner shown in Figure 1a. The laser of the probe scans the surface from the vertical direction, denoted as the $y$ direction, and the height obtained from the scanning corresponds to the vertical height of the point where the laser is located. However, when there is an abrasive mark on the scanning path, as represented by the enlarged abrasive mark in Figure 1b, the laser light emitted by the 3D profiler's probe is represented by the arrow $y$ on the left side of the figure. The depth of the abrasive mark obtained by the 3D profiler at this point is the height of the point relative to the reference surface in the $y$ direction, denoted as $p$. In order to accurately measure the true depth of the abrasive mark, the laser light must be emitted from the radial direction, specifically the $x$ direction of the surface towards the abrasive mark. The depth of the abrasive mark is denoted as $q$. When no abrasive mark is present, the height obtained from the scanning corresponds to the vertical height of the point, and the depth of the abrasive mark is effectively zero.

However, in the presence of an abrasive mark, $p$ and $q$ are not equal. Therefore, in order to reconstruct a more accurate plane, it is necessary to use the height data represented by $p$.

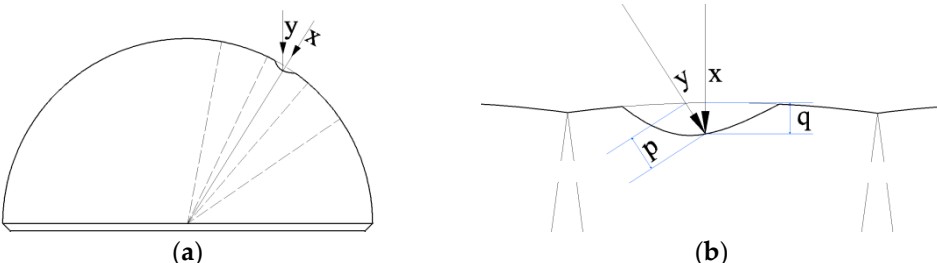

**Figure 1.** Scanning mode analysis. (**a**) Scanning light. (**b**) Magnified abrasion marks.

Let a three-dimensional surface satisfy the expression of $z = f(x, y)$. The origin position of the function coincides with the coordinates of the center of the surface to be measured $(x_0, y_0, z_0)$, and the step sizes in the $x$, $y$ direction set during the process of three-dimensional morphology scanning are $\Delta x$ and $\Delta y$. The height $z_{i,j}$ in the expression at this time, after $i, j$ moves in the $x$, $y$ direction, is:

$$z_{i,j} = f(x_0 \pm i\Delta x,\ y_0 \pm j\Delta y) \tag{1}$$

From this, it is possible to calculate the distance of all measured positions from the center of the regular surface in the radial direction:

$$t_{i,j} = \sqrt{(x_0 \pm i\Delta x)^2 + (y_0 \pm j\Delta y)^2 + z_{i,j}^2} \tag{2}$$

where $z_{i,j}$ is the theoretical height at any measured position on the surface equation.

The coordinates obtained from scanning are subjected to a coordinate transformation so that the center of the coordinates coincides with the center of the surface, resulting in the coordinates $\left(x \pm i\Delta x,\ y \pm j\Delta y,\ z'_{i,j}\right)$ and the radial distance

$$t'_{i,j} = \sqrt{(x \pm i\Delta x)^2 + (y \pm j\Delta y)^2 + (z'_{i,j})^2} \tag{3}$$

After reconstruction with the depth of each measurement position as the height, the height $h_{i,j}$ of each measurement position on the plane is

$$h_{i,j} = t'_{i,j} - t_{i,j} \tag{4}$$

In addition, note that the slope of each scan point is

$$k = \tan\alpha = \frac{z}{\sqrt{x^2 + y^2}} \tag{5}$$

Expanding the height $h_{i,j}$ of the measured position to the horizontal plane, there are

$$H'_0 = h_{i,j} / \cos\alpha = h_{i,j} \frac{\sqrt{f^2(x, y) + x^2 + y^2}}{\sqrt{x^2 + y^2}} \tag{6}$$

Taking the sphere as the object of study, its equation is $z = f(x, y) = \sqrt{R^2 - x^2 - y^2}$, where $R$ is the radius of the sphere, and the coordinates of the center point of the scan are $(x_0, y_0, z_0)$. Since the distance from the center of the sphere to any point on the surface is equal to $R$, the radial distances from each point on the surface are $R$.

The height matrix $H'_0$, formed by the height difference $h_{i,j} = t'_{i,j} - R$ between any point in the plane coordinates and the center of the scan, is

$$H'_0 = h_{i,j}/\cos\alpha = (t'_{i,j} - R)\frac{R}{\sqrt{x^2 + y^2}} \tag{7}$$

The reconstructed 3D shape can be obtained by inserting the corresponding $x$, $y$ coordinates into each element of the above height matrix $H'_0$ and plotting the 3D surface.

## 3. Experimental Test

In this paper, the authors perform validation work to verify the effectiveness of the proposed reconstruction method. We select the bottom pivot spherical bearing friction pair simulation specimen of the Three Gorges Dam as the test object. The surface morphology of the spherical bearing friction pair is measured using the NANOVEA ST400 three-dimensional surface profiler. Additionally, a scaled-down model (1:20) of the Three Gorges Dam pivot spherical bearing friction pair is scanned using the same profiling technique. The data obtained from these scanning processes are considered as the measured data for the validation. To validate the reconstruction method, the authors process the measured data and generate corresponding three-dimensional morphology graphs using MATLAB. By comparing the reconstructed surface with the actual surface of the test object, the authors can assess the accuracy and reliability of the proposed method. The validation process helps in evaluating the performance of the reconstruction method and provides evidence for its applicability in real-world scenarios, such as the Three Gorges Dam pivot.

### 3.1. Acquisition of Actual Measurement Data

The selected specimens for the spherical bearing friction pair simulation consist of two parts, the axial tile and the mushroom head. These two parts are shown in Figure 2, where the left figure represents the axial tile and the right figure represents the mushroom head. The materials used for the preparation of these specimens are tin bronze ZQSn6-6-3 alloy and 45# steel. The axial tile features three grooves on its inner side, which serve as the grease filling positions. The dimensions of the oil grooves differ for specimen 1 and specimen 2. For specimen 1, the width and depth of the oil grooves in the axial tile are 2 mm and 1.00 mm, respectively. For specimen 2, the width and depth of the oil grooves in the axial tile are 3 mm and 0.25 mm, respectively. The mushroom head has a top radius of 25 mm. Its surface undergoes quenching, which increases its hardness to HRC45~48. The fit tolerance between the mushroom head and the axial tile is specified as $H7/f6$.

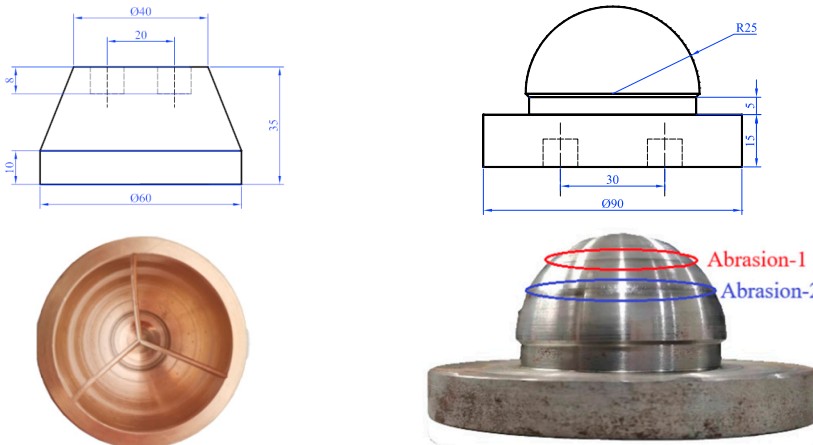

**Figure 2.** Miter gate bottom pivot simulation test piece: bearing bush (**left**) and mushroom head (**right**). (Unit: mm).

The test is conducted using a self-developed spherical wear tester, as depicted in Figure 3. The tester is set with parameters including a rotational speed of 5 revolutions per minute (5 r/min) and a load of 1500 N; the lubrication condition is grease lubrication. Specifically, 3# lithium grease is used as the lubricant for the test. Each individual test lasts for 5 h, and a total of 25 h of testing are completed. At the conclusion of the test, the surface of the specimens is wiped to remove the grease, followed by cleaning with an ultrasonic cleaner to ensure the removal of any remaining grease; anhydrous ethanol $C_2H_5OH$ is used as the cleaning solvent. In this test, the volumetric method was used to measure the wear of the axial tile. A beaker is filled with a certain mass of water and placed on a balance to measure its mass ($m_1$); the inner surface of the axial tile before wear is filled with water, the mass of water remaining in the beaker is measured ($m_2$), and the volume of the inner surface of the axial tile before wear is $V_1 = (m_1 - m_2)/\rho_{\text{water}}$. The same method is used to measure the volume of the worn-out axial tile after wear; the volume of the worn out axial tile is calculated as $V_2 - V_1$, which allows the wear volume of the axial tile to be calculated as $m = \rho_{\text{copper}}(V_2 - V_1)$. Specimen 1 had a wear of 0.1 mm and specimen 2 had a wear of 0.08 mm. Through conversion, it is determined that the wear volume of the axial tile in sample 1 is greater than that of sample 2. Moreover, the change in wear volume between the friction pair is similar. Consequently, it can be deduced that the wear volume of the mushroom head in specimen 1 is larger than that of specimen 2.

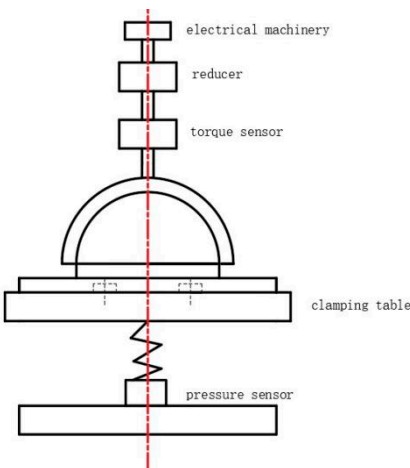

**Figure 3.** Friction wear testing machine and the clamping way.

To establish a benchmark for plane reconstruction, a 3D surface morphology instrument is utilized to scan the mushroom head after the test. The scanning process begins with the top of the mushroom head, covering an area of 14 mm × 14 mm × 3 mm. The resulting 3D morphology is presented in Figure 4a. For a closer view, Figure 4b displays an enlarged image of the top section. A noticeable "bump" can be observed at the highest point in the three-dimensional morphology, and the phenomenon is attributed to the presence of lubrication structures in the axial tile. The matching parts of the mushroom head's top and the axial tile do not make direct contact. Instead, the contact is facilitated by the presence of grease, where there is no relative motion between the two surfaces. Consequently, no wear occurs in this region. However, the surrounding areas of the mushroom head are in direct contact with the axial tile. As a result, a certain amount of wear is observed in these areas.

Based on the positional relationship between the "bump" and the scanning center, the three-dimensional coordinates obtained from the 3D morphology can be adjusted to determine the positional relationship between the mushroom head's center and the scanning center. This allows for the alignment of the measured positional coordinates, ensuring that the zero point of the coordinates coincides with the center of the sphere. Such adjustment is crucial for the subsequent planar reconstruction of the sphere with the assistance of the "bump" feature. After applying the "to the center" operation described above, the two wear marks on the mushroom head resulting from the friction wear test are

scanned, as depicted in Figure 2. The top wear mark is denoted as "abrasion mark 1", while the bottom wear mark is referred to as "abrasion mark 2". Due to the height limitations of the scanning instrument, only a portion of the wear mark region can be selected for scanning. For each group of specimens, the scanning parameters for the 3D morphology instrument are set to 5 mm × 2 mm × 3 mm. The step size in the $x$ and $y$ directions is set to 5 μm. The initial 3D morphology of each group of specimens is illustrated in Figure 4c–f, where (c) shows the abrasion mark 1 of specimen 1, (d) shows the abrasion mark 2 of specimen 1, (e) shows the abrasion mark 1 of specimen 2, and (f) shows the abrasion mark 2 of specimen 2.

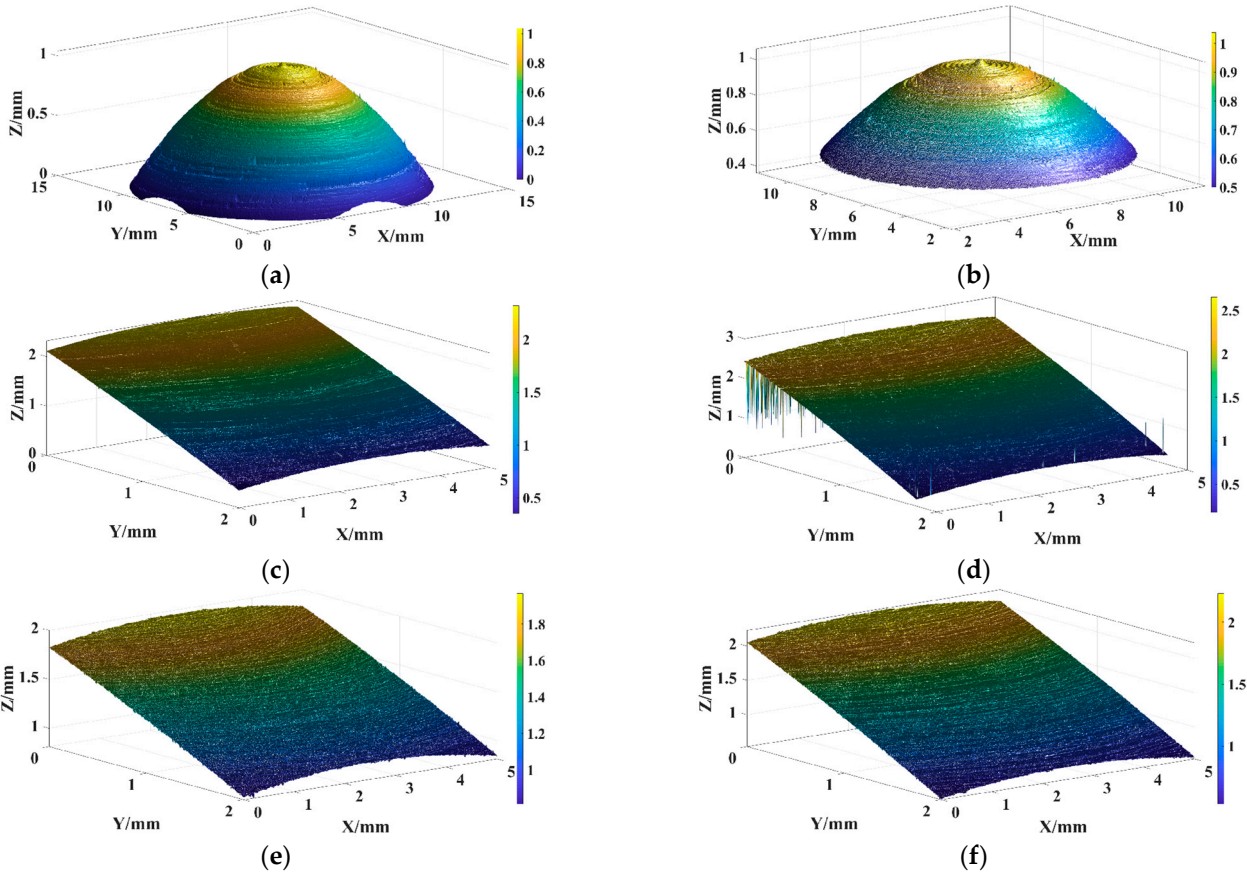

**Figure 4.** Initial three-dimensional morphology. (**a**) Top three-dimensional morphology. (**b**) Local three-dimensional morphology. (**c**) Specimen 1 abrasion mark 1. (**d**) Specimen 1 abrasion mark 2. (**e**) Specimen 2 abrasion mark 1. (**f**) Specimen 2 abrasion mark 2.

### 3.2. Three-Dimensional Reconstruction

From the structure diagram of the mushroom head, we can know the radius of the sphere $R = 25$ mm, set the coordinates of the center of the sphere as $(0, 0, 0)$; the equation of the surface is $x^2 + y^2 + z^2 = 625$, $z > 0$. However, since the center of the scanning area of the 3D profiler is not the center of the sphere, it is necessary to make use of the positional relationship between the center of the scanning area and the center of the spherical surface to carry out a simple coordinate transformation of the above equation to make the transformed coordinates of the $(x', y', z')$ origin coincide with the center of the ruled surface. We then need to substitute $x, y$ in the data from the scanning area into the equation to obtain the theoretical height Z on the ruled surface, so the distance from the surface center to each measurement position can be calculated. Since the selected surface is a sphere, the distance from each point on the sphere to the center is constant and equal to $R = 25$ mm, so the height difference $h = \sqrt{x'^2 + y'^2 + z'^2} - 25$ at each point needs to be

calculated, the unit is mm, and then $x, y, h$ is used as the $x, y, z$-axis, respectively, to draw the corresponding three-dimensional morphology diagrams.

Figure 5 depicts the reconstructed 3D morphology of the specimens. Specifically, (a) and (b) represent the reconstructed 3D morphology of sample 1 at abrasion mark 1 and abrasion mark 2, respectively. Similarly, (c) and (d) show the reconstructed 3D morphology of sample 2 at abrasion mark 1 and abrasion mark 2, respectively. The depth of the indentations in the figures indicates the severity of wear, with deeper indentations representing more significant wear. It is evident from the figures that the wear at the abrasion marks is more severe compared to the surrounding areas. The values of the specific wear depths can be seen in Figure 5. In sample 1, the maximum depth of abrasion at abrasion mark 1 is 0.05 mm, while at abrasion mark 2, it is 0.03 mm. For sample 2, the maximum depth of abrasion at abrasion mark 1 is 0.03 mm, whereas at abrasion mark 2, it is 0.02 mm. These measurements highlight that sample 1 exhibits more severe abrasion compared to sample 2, with the most significant wear occurring at abrasion mark 1, aligning with the wear pattern observed on the mushroom head. According to the research literature [14], the wear on the mushroom head is greatest at its maximum stress point. Taking into account the weight, gate leaf, and gate width of the Three Gorges Dam miter gate, it is known that the maximum stress occurs at an approximate angle of 60° from the horizontal plane. The proximity of wear mark 1 to this position indicates that it corresponds to the area experiencing the highest stress.

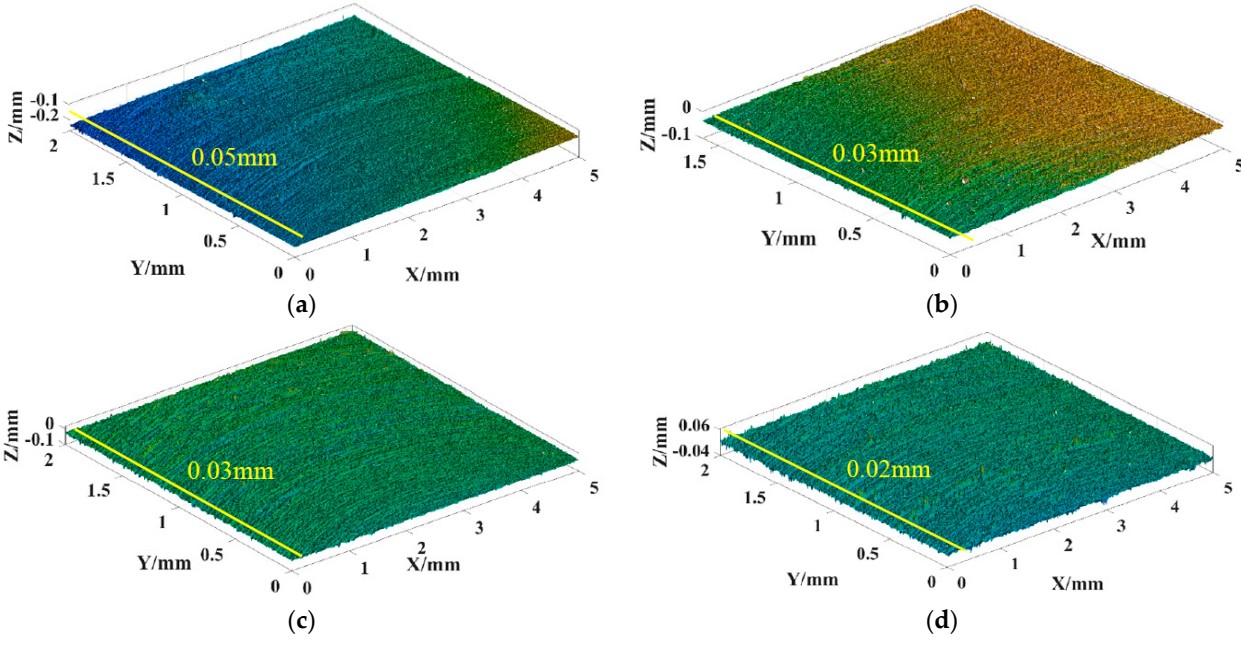

**Figure 5.** Reconstruction of three-dimensional morphology. (**a**) Specimen 1 abrasion mark 1. (**b**) Specimen 1 abrasion mark 2. (**c**) Specimen 2 abrasion mark 1. (**d**) Specimen 2 abrasion mark 2.

### 3.3. Wear of Different Parts of Mushroom Head

From the physical image of the mushroom head in Figure 2, it can be seen that there are two abrasion marks of different depths in the scanned area, the abrasion mark near the top of the sphere (inside the red ellipse), abrasion mark 1, is lighter, and the abrasion mark away from the top of the sphere (inside the blue ellipse), abrasion mark 2, is deeper. The abrasion study was carried out on the above reconstructed images of specimen 1 and specimen 2, and the obtained 3D morphology was evaluated using the following parameters, which are the contour arithmetic mean deviation $Sa$, contour root mean square deviation $Sq$, and crag $Sku$.

$$Sa = \frac{1}{A}\iint\limits_{A}|Z(x, y)|dxdy \tag{8}$$

$$Sq = \sqrt{\frac{1}{A}\iint\limits_A Z^2(x, y)dxdy} \tag{9}$$

$$Sku = \frac{1}{Sq^4}[\frac{1}{A}\iint\limits_A Z^4(x, y)dxdy] \tag{10}$$

where $A$ is the area of the measurement area and $Z(x, y)$ are all the distances from points in the area to the reference plane.

### 3.3.1. Analysis of Wear and Tear of Specimen 1

The area 2.1 mm $\leq x \leq$ 2.9 mm in Figure 5a,b is selected as the object of analysis. The area is divided in the *y*-axis direction according to 0.8 mm segments and surface characterization is carried out. All 3D morphology obtained is shown in Figure 6.

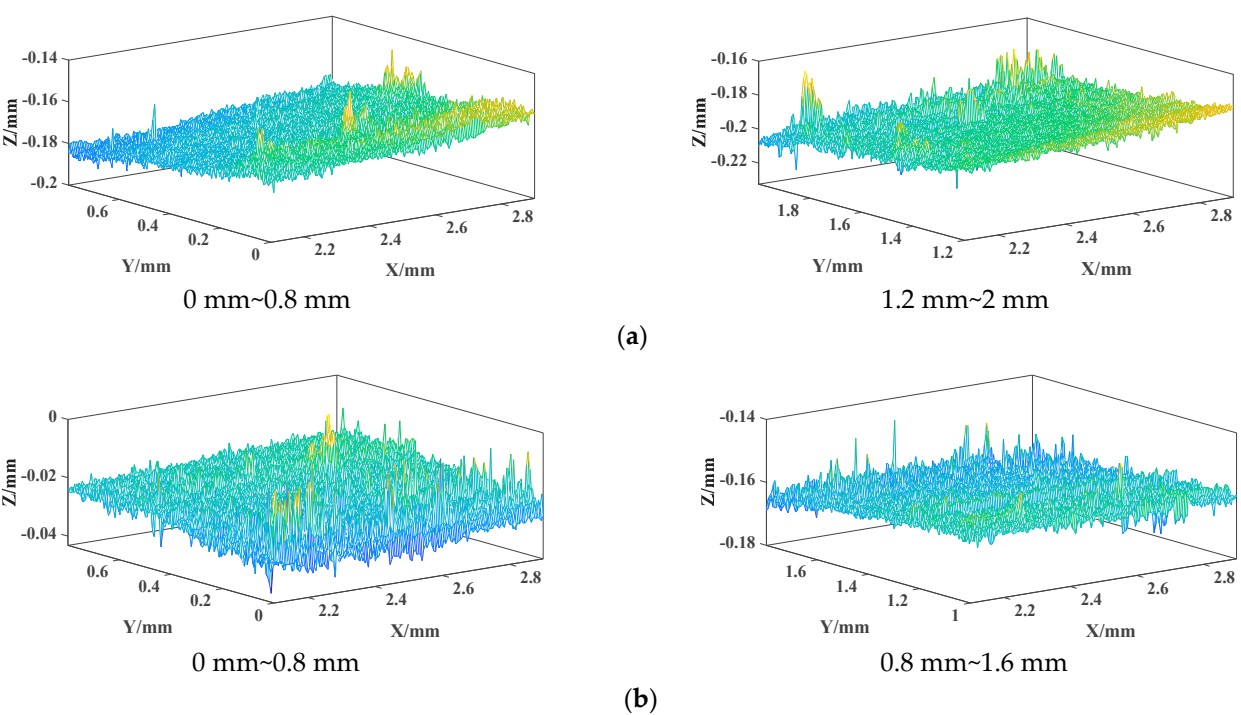

**Figure 6.** Three-dimensional morphology for some parts of specimen 1. (**a**) Three-dimensional morphology of each part of abrasion mark 1. (**b**) Three-dimensional morphology of each part of abrasion mark 2.

As can be seen from the 3D morphology in Figure 6, the surface of the two regions of abrasion mark 2 is much flatter, and the abrasion amount of abrasion mark 1 is much larger. The results are shown in Table 1. The average deviation *Sa* and the root mean square deviation *Sq* of the surface contour of wear mark 2 are much smaller than that of wear mark 1, which indicates that the flatness of wear mark 2 is better, but the average deviation *Sa* and the root mean square deviation *Sq* of the surface contour of the two wear marks are very small, so the friction partner can still work normally. The value of the cliff *Sku* is mostly larger than 3 mm, which indicates that the distribution of the surface height is as sharp as a needle, and the surface lubrication film is not particularly complete. The *Sku* of the area of 0~0.8 mm in wear mark 1 is less than 3 mm, which means that the distribution of the height is relatively even, because it is located near the top of the sphere, which is far away from the position of maximum stress.

**Table 1.** Value of *Sa*, *Sq*, and *Sku* for every part in the area of specimen 1.

| Item | Area/mm | *Sa*/μm | *Sq*/μm | *Sku* |
|---|---|---|---|---|
| abrasion mark 1 | 0~0.8 | 4.4 | 5.5 | 2.5729 |
| | 1.2~2 | 4.6 | 5.8 | 3.4420 |
| abrasion mark 2 | 0~0.8 | 2.5 | 3.3 | 4.8997 |
| | 0.8~1.6 | 1.6 | 2.1 | 7.7632 |

3.3.2. Analysis of Wear and Tear of Specimen 2

Taking 3.5 mm < *y* < 4.3 mm as the analyzed area in Figure 5c,d, the 3D morphology is also intercepted according to a section of 0.8 mm, and the 3D morphology of each part of abrasion mark 1 and abrasion mark 2 is obtained as shown in Figure 7.

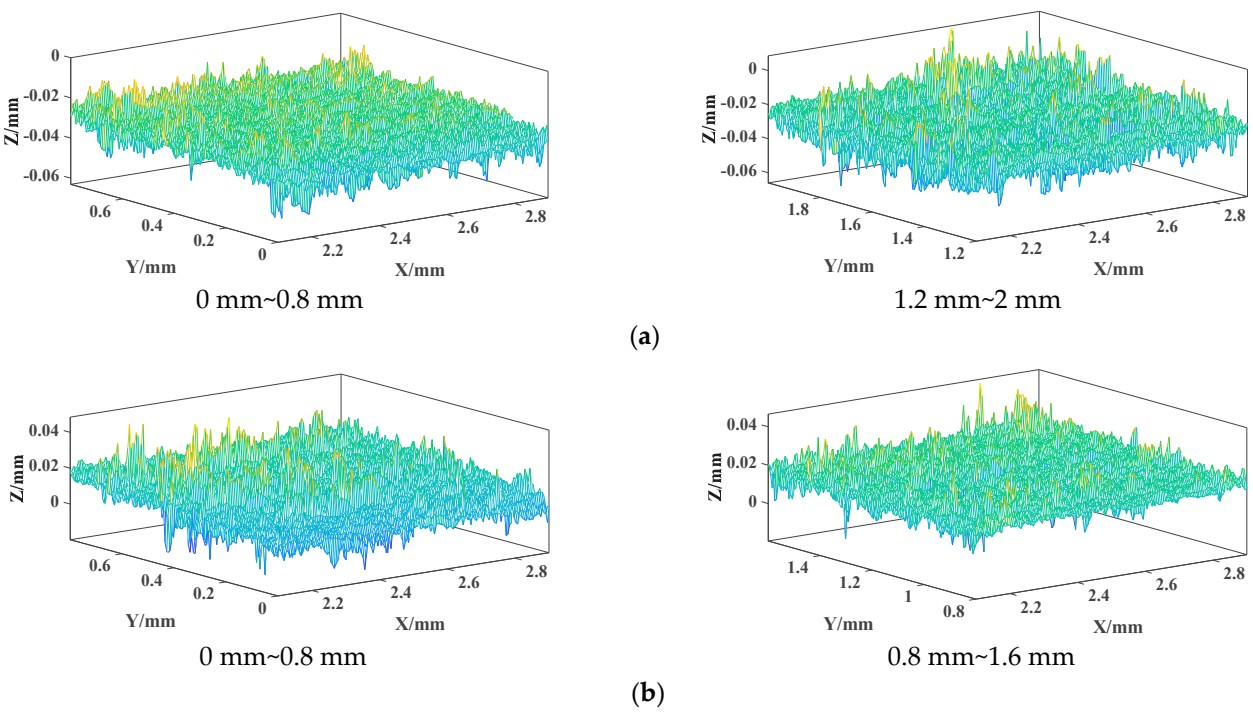

**Figure 7.** Three-dimensional morphology for some parts of specimen 2. (**a**) Three-dimensional morphology of each part of abrasion mark−1. (**b**) Three-dimensional morphology of each part of abrasion mark−2.

Table 2 displays the statistical parameters pertaining to the 3D morphology of each component of sample 2. Observing the values of the arithmetic mean deviation of the contour *Sa* in Table 2 and the three-dimensional morphology of the 1.2 mm to 2 mm region in abrasion mark 1, it can be seen that the arithmetic mean deviation of the contour is smaller than 4.2 μm, which indicates that the surface is relatively smooth and the integrity of the lubrication film is relatively high. Moreover, the degree of cragginess *Sku* is greater than 3 mm, suggesting that the surface spikes are sharp. By examining the image information, it can be inferred that there are numerous spikes present on the surface contour. However, their distribution is uniform, leading to minimal variations in the arithmetic mean deviation and the root mean square deviation. Conversely, the cragginess appears to be significant. Therefore, considering the image information as well, it is evident that there are several evenly distributed surface spikes, resulting in very small arithmetic mean deviation (*Sa*) and root mean square deviation (*Sq*), while exhibiting a substantial degree of steepness (*Sku*).

**Table 2.** Value of *Sa*, *Sq*, and *Sku* for every part in the area of specimen 2.

| Item | Area/mm | *Sa*/μm | *Sq*/μm | *Sku* |
|---|---|---|---|---|
| abrasion mark 1 | 0~0.8 | 3.0 | 4.1 | 5.8810 |
| | 1.2~2 | 4.2 | 6.0 | 5.8361 |
| abrasion mark 2 | 0~0.8 | 3.7 | 5.1 | 6.9660 |
| | 0.8~1.6 | 2.4 | 3.6 | 12.5155 |

3.3.3. Reconstruction Results and Accuracy Analysis

As a result, both specimen 1 and specimen 2 have the roughest abrasion marks near the maximum stress point (the area of 1.2~2 mm at abrasion mark 1 of each specimen), and the arithmetic mean deviation of the surface profile *Sa* reaches a maximum of 4.6 μm, with the largest abrasion at abrasion mark 1 and the smallest abrasion at abrasion mark 2 in specimen 1. The abrasion of specimen 1 is extremely uneven in the course of the test, and the abrasion of specimen 2 is relatively more uniform. The measurement indexes of the two abrasion marks are close to each other.

Although the reliability of the reconstructed plane height decreases, the relative positional relationship between the points on the reconstructed plane does not change, so this error will not affect the acquisition of the reconstructed plane parameters. Secondly, because the surface of the untested part is not exactly a regular surface, the distribution of micro-convex bodies on the surface will have a certain impact on the results, and therefore there is a certain degree of systematic error. Moreover, the reconstruction is aimed at the direction of the projection to the surface "expansion", but the real surface for the surface "expansion" is not the same. In addition, the reconstruction aims to "unfold" the surface in the projection direction, but the real way of unfolding the surface should be to unfold the surface radially according to the area at each measurement point. In this paper, the total measurement area obtained by unfolding the surface in the projection direction has a maximum difference of 11% compared with that obtained by unfolding the surface in the radial direction. Therefore, there is a certain degree of error in the area of the planar reconstruction, but the surface parameters can be extracted from the surface because we can calculate it using the coordinates of each point. However, in the extraction of surface feature parameters, the reliability of the information obtained from the reconstructed plane can be guaranteed because the arithmetic mean deviation of the contour, the root mean square deviation of the contour, and other surface morphology parameters can be calculated using the coordinates of each point.

**4. Conclusions**

In this paper, a simple and effective reconstruction method is proposed for regular surfaces, which has considerable application value in engineering, especially in the online measurement of the three-dimensional morphology of regular surfaces.

1. Taking a spherical surface as an example, from the geometry of the test object it is possible to obtain the equation of the spherical surface, design a suitable algorithm, and then carry out the test object scanning calculation, on the premise of ignoring the error introduced by the three-dimensional morphology instrument after the calculation of the reconstructed three-dimensional morphology.
2. After reconstruction, the maximum depth of abrasion at abrasion mark 1 of sample 1 is 0.05 mm, the maximum depth of abrasion at abrasion mark 2 is 0.03 mm, the maximum depth of abrasion at abrasion mark 1 of sample 2 is 0.03 mm, and the maximum depth of abrasion at abrasion mark 2 is 0.02 mm. The maximum arithmetic mean deviation of the contour *Sa* is 4.6 μm, which appears at the position of abrasion mark 1 in sample 1. The maximum value of *Sq* is 6.0 μm, which appears at the position of wear mark 1 in specimen 2. Both of these are located at the position of maximum stress of the bottom pivot friction pair, which coincides with the actual wear law. The surface characteristics are more clearly expressed in the reconstructed

plane, and the change of the depth of the wear marks is more intuitive. At the same time, from the reconstruction principle, it can be seen that the height of each point on the image represents the depth of wear, so the volume between the reconstructed three-dimensional surface and the $Z = 0$ plane can be regarded as the wear volume of the scanning range, and the wear volume can be obtained after the conversion. Due to the special characteristics of the water environment in which the Three Gorges gate is situated, it is of great reference significance for the measurement of wear and tear in engineering practice and for the subsequent study of tribology.

**Author Contributions:** Conceptualization, X.X. and Z.F.; methodology, N.X.; software, Z.Z.; validation, Z.F., X.X. and X.Z.; formal analysis, Z.Z.; investigation, Z.F.; resources, X.X.; data curation, Z.F.; writing—original draft preparation, Z.F.; writing—review and editing, X.X.; visualization, X.Z.; supervision, X.X.; project administration, X.Z.; funding acquisition, X.X. All authors have read and agreed to the published version of the manuscript.

**Funding:** This research was carried funded by the National Nature Science Foundation of China (No. 52175177) (Study on Solid Lubrication Wear Mechanism and Wear Characterization Method of Large Sliding Spherical Friction Pairs, 2022.01–2025.12).

**Acknowledgments:** This research was carried out with the National Nature Science Foundation of China (No. 52175177) (Study on Solid Lubrication Wear Mechanism and Wear Characterization Method of Large Sliding Spherical Friction Pairs, 2022.01–2025.12).

**Conflicts of Interest:** The authors declare no conflict of interest.

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
