# Peer review of "Planar Reconstruction of Regular Surfaces’ Three-Dimensional Morphology and Tribology Application"

_lubricants, doi:10.3390/lubricants11090398_

Round 1
Reviewer 1 Report
Overall comment:
The manuscript entitled “Planar Reconstruction of Regular Surfaces’ Three-dimensional Morphology and Tribology Application”, by Xu et al., reports an approach to construct regular surface equations to normalize 3D morphologies. Here, the model is tested with wear data. In my view, this paper shows an interesting approach to reconstructing 3D surfaces, and the mathematical factors and considerations seem to be solid enough and the results corroborate some assumptions. I have some concerns about how the results are reported and, below, you may find some suggestions, questions, and recommendations to improve these points. I think that solving these and other issues (possibly stated by both the Editor and reviewers) may have a decisive impact on the relevance and applicability of the presented model. Once the approach is clearer, it would be more tested and, finally, widely used to help solve real-world problems. I recommend a revision before acceptance and I’m looking forward to receiving a revised, new version of the manuscript.
Some comments/suggestions/concerns:
- First, I recommend a total review of the captions of the figures. The figures should be as independent as possible of the main text, it is: The figure + caption system may lead the reader to a full understanding of what is being presented.
- Figure 2: You should inform the units of the dimensions (in the figure or the caption or both).
- Figure 6: It is very difficult to follow this figure. The subplots are not easily identified.
- The methods section is missing and such information is extremely mixed with the results. It is difficult to follow the results with successive stops to a new part of the methodology. This kind of confuses the reader. Moreover, some information is missing, for example:
a) The ultrasonic cleaning was conducted with what kind of solvent?
b) Has the known Archimedes drainage followed a published protocol? All the protocols should be referenced or otherwise fully described.
c) "Through conversion, it is determined that the wear volume [...]". What conversion?
- "Based on the ferrography analysis reported in literature reference [14], it is observed that the predominant wear type between the friction pair is adhesive wear." Did the authors analyze the current system or are talking about results obtained elsewhere? It is not clear.
- I did not understand one point: on page 4 of the PDF file: "The surface morphology of the bottom pivot is measured using the NANOVEA ST400 three-dimensional surface profiler. Additionally, a scaled-down model (1:20) of the Three Gorges Dam pivot mushroom head is scanned using the same profiling technique." Two specimens were tested? Is the "spherical bearing friction pair simulation" in section 3.1 another one?
- "contour are both smaller than 4.2μm. This indicates that the surface is relatively smooth, and the lubrication film's integrity is relatively high". Is this assumption based on any reference? Every comparison should be based on specific data, otherwise, it is not clear what is the standard.
- Why the Sku factor is much more precise (four digits) than the Sa and Sq in both Tables 1 and 2? Is it really that precise?
Author Response
Detailed Response to Reviewers
Dear Editor and Reviewers :
Thank you for your letter and for the reviewers’ comments concerning our manuscript entitled "Planar Reconstruction of Regular Surfaces’ Three-dimensional Morphology and Tribology Application" (2598675) . Those comments are all valuable and very helpful for revising and improving our paper, as well as the important guiding significance to our researches. We have studied comments carefully and have made correction which we hope meet with approval. Revised portion are marked in yellow in the paper. The main corrections in the paper and the responds to the reviewer’s comments are as flowing:
- First, I recommend a total review of the captions of the figures. The figures should be as independent as possible of the main text, it is: The figure + caption system may lead the reader to a full understanding of what is being presented.
Thank you for your advice. We have followed up on your suggestions and made changes.
- Figure 2: You should inform the units of the dimensions (in the figure or the caption or both).
We are sorry for our carelessness. In our resubmitted manuscript, we have added it.
- Figure 6: It is very difficult to follow this figure. The subplots are not easily identified.
The use of three-dimensional topography to find the highest point of the specimen, when the corresponding coordinate point can be considered as the reference value of (x0,y0,z0), and then adjust the probe position of the three-dimensional topography, corresponding to the specimen's abrasion marks, can be displayed through the three-dimensional topography readings and set up scanning area, to get the other parameters of Formula 8-10. On the basis of the basic data in Fig. 4, the data in Fig. 6 are obtained through MATLAB calculation.
- The methods section is missing and such information is extremely mixed with the results. It is difficult to follow the results with successive stops to a new part of the methodology. This kind of confuses the reader. Moreover, some information is missing, for example:
- The ultrasonic cleaning was conducted with what kind of solvent?
Thank you for your advice. In our resubmitted manuscript, we have added anhydrous ethanol C2H5OH is used as cleaning solvent.
- Has the known Archimedes drainage followed a published protocol? All the protocols should be referenced or otherwise fully described.、
Because it is difficult to measure the amount of wear and tear, the group has carried out a series of studies, such as the indentation method, weighing method, etc., and found that it is relatively accurate to measure by the volumetric method through comparison. This is, of course, one of the more classical methods in tribology.
- "Through conversion, it is determined that the wear volume [...]". What conversion?
In this test, the volumetric method was used to measure the wear of the axial tile. A beaker is filled with a certain mass of water and placed on a balance to measure its mass m1, the inner surface of the axial tile before wear is filled with water, the mass of water remaining in the beaker is measured m2, the volume of the inner surface of the axial tile before wear is V1=(m1-m2)/ρwater. The same method is used to measure the volume of the worn out axial tile after wear is measured and the volume of the worn out axial tile is calculated to be V2-V1, which is able to calculate the wear volume of the axial tile to be m=ρcopper(V2-V1).
- "Based on the ferrography analysis reported in literature reference [14], it is observed that the predominant wear type between the friction pair is adhesive wear." Did the authors analyze the current system or are talking about results obtained elsewhere? It is not clear.
In our previous study, we originally intended to correspond to the wear mechanism, and considered the option of deleting it as it was not very relevant to the main idea of the article.
- I did not understand one point: on page 4 of the PDF file: "The surface morphology of the bottom pivot is measured using the NANOVEA ST400 three-dimensional surface profiler. Additionally, a scaled-down model (1:20) of the Three Gorges Dam pivot mushroom head is scanned using the same profiling technique." Two specimens were tested? Is the "spherical bearing friction pair simulation" in section 3.1 another one?
The bottom pivot of the gate of the Three Gorges Dam consists of a pair of friction pairs of mushroom head and shaft tile, which are also a pair of spherical bearing friction pairs. After the modification, the spherical bearing friction pair is used as a unified description.
- "contour are both smaller than 4.2μm. This indicates that the surface is relatively smooth, and the lubrication film's integrity is relatively high". Is this assumption based on any reference? Every comparison should be based on specific data, otherwise, it is not clear what is the standard.
We were sorry for our careless mistakes. The part of the paragraph that was not clear has been corrected. The definition of this parameter is derived from a combination of the values of the average deviation of the surface profile Sa in the region of specimen 2 in Table 2 and the three-dimensional morphology of the 1.2 mm to 2 mm region of the abrasion mark 1 in Fig. 7, which has a maximum value of only 4.2μm and is presented as smooth in the three-dimensional morphology diagrams, the authors concluded that the lubrication film integrity is high.
- Why the Sku factor is much more precise (four digits) than the Sa and Sq in both Tables 1 and 2? Is it really that precise?
Sa and Sq are the surface arithmetic mean and root-mean-square values, respectively, calculated in mm. In the table the unit has been changed to μm, and Sku is the cragness with no unit, so Sa and Sq are shown as one digit when they are also accurate to four digits.

Reviewer 2 Report
1. The authorship label sequence number 1, 2, 3 and a, b, c should be unified.
2. The author mentioned in Section 3.1 that "The wear volume of the axial tiles is then measured using the Archimedes drainage method. Through conversion, it is determined that the wear volume of the axial tile in sample 1 is greater than that of sample 2. The specific wear volume or wear amount of two groups of samples is given for comparison.
3. The format of some formulas needs attention, such as formula (1)-(7): the formula needs to be centered, the number is right-aligned, and other same details need to be paid attention to.
4. In Figure 4 c, d, e, f, the author gave the initial three-dimensional morphology of each group of specimens but did not give a proper explanation. Please explain.
5. The author pointed out in Figure 5: "The depth of the indentations in the figures indicates the severity of wear, with deeper indentations representing more significant wear. It is evident from the figures that the wear at the abrasion marks is more severe compared to the surrounding areas ". However, the specific indentation depth cannot be seen intuitively in the figure; please indicate it in the figure.
6. In Figure 6, the author gives Three-dimensional morphology of each part of abrasion mark-1 and mark-2, but the arrangement of the pictures is less organized, which is not conducive to readers' understanding and analysis. Please improve.
Minor revisions are needed
Author Response
Detailed Response to Reviewers
Dear Editor and Reviewers :
Thank you for your letter and for the reviewers’ comments concerning our manuscript entitled "Planar Reconstruction of Regular Surfaces’ Three-dimensional Morphology and Tribology Application" (2598675) . Those comments are all valuable and very helpful for revising and improving our paper, as well as the important guiding significance to our researches. We have studied comments carefully and have made correction which we hope meet with approval. Revised portion are marked in yellow in the paper. The main corrections in the paper and the responds to the reviewer’s comments are as flowing:
Comments and Suggestions for Authors
- The authorship label sequence number 1, 2, 3 and a, b, c should be unified.
Thank you for your advice. We have followed up on your suggestions and made changes.
We were sorry for our careless mistakes.
- The author mentioned in Section 3.1 that "The wear volume of the axial tiles is then measured using the Archimedes drainage method. Through conversion, it is determined that the wear volume of the axial tile in sample 1 is greater than that of sample 2. The specific wear volume or wear amount of two groups of samples is given for comparison.
In this test, the volumetric method was used to measure the wear of the axial tile. A beaker is filled with a certain mass of water and placed on a balance to measure its mass m1, the inner surface of the axial tile before wear is filled with water, the mass of water remaining in the beaker is measured m2, the volume of the inner surface of the axial tile before wear is V1=(m1-m2)/ρwater. The same method is used to measure the volume of the worn out axial tile after wear is measured and the volume of the worn out axial tile is calculated to be V2-V1, which is able to calculate the wear volume of the axial tile to be m=ρcopper(V2-V1). Specimen 1 had a wear of 0.1mm and specimen 2 had a wear of 0.08mm.
- The format of some formulas needs attention, such as formula (1)-(7): the formula needs to be centered, the number is right-aligned, and other same details need to be paid attention to.
Thank you for your advice. We have made corrections. We are sorry for our carelessness.
- In Figure 4 c, d, e, f, the author gave the initial three-dimensional morphology of each group of specimens but did not give a proper explanation. Please explain.
Thank you for your advice. We have made corrections. In our resubmitted manuscript,we have explained it. Where Figure 4 (c) shows the abrasion mark 1 of specimen 1, (d) shows the abrasion mark 2 of specimen 1, (e) shows the abrasion mark 1 of specimen 2, (f) shows the abrasion mark 2 of specimen 2.
- The author pointed out in Figure 5: "The depth of the indentations in the figures indicates the severity of wear, with deeper indentations representing more significant wear. It is evident from the figures that the wear at the abrasion marks is more severe compared to the surrounding areas ". However, the specific indentation depth cannot be seen intuitively in the figure; please indicate it in the figure.
Thank you for your advice. We have made corrections. The values of the specific wear depths can be seen in Figure 5.
- In Figure 6, the author gives Three-dimensional morphology of each part of abrasion mark-1 and mark-2, but the arrangement of the pictures is less organized, which is not conducive to readers' understanding and analysis. Please improve.
Thank you for your advice. We have made corrections. Due to the height limitations of the scanning instrument, only a portion of the wear mark region can be selected for scanning. For each group of specimens, the scanning parameters for the 3D morphology instrument are set to 5mm×2mm×3mm. The step size in the x and y directions is set to 5μm. The authors believe that combining the location of wear in the pictures with the data in the tables can be well presented as well as easily analyzed.

Round 2
Reviewer 1 Report
I am satisfied with the answer, the current version of the manuscript, and the fact that the authors followed the comments. In my opinion, the manuscript is suitable for acceptance. The quality of the presentation of the results was significantly improved, resulting in the enhancement of the whole study. With all this in mind, I recommend the publication.